# Skin and Arterial Wall Deposits of 18F-NaF and Severity of Disease in Patients with Pseudoxanthoma Elasticum

**DOI:** 10.3390/jcm9051393

**Published:** 2020-05-08

**Authors:** Antonio Gutierrez-Cardo, Eugenia Lillo, Belén Murcia-Casas, Juan Luis Carrillo-Linares, Francisco García-Argüello, Purificación Sánchez-Sánchez, Alejandro Rodriguez-Morata, Isabel Baquero Aranda, Miguel Ángel Sánchez-Chaparro, María García-Fernández, Pedro Valdivielso

**Affiliations:** 1Molecular Image Unit, Nuclear Medicine, Fundación General de la Universidad de Málaga, 29010 Málaga, Spain; algutierrezc@icloud.com (A.G.-C.); elillo@fguma.es (E.L.); 2Internal Medicine Unit, Hospital Universitario Virgen de la Victoria, 29010 Málaga, Spain; belenmurciacasas9@outlook.es (B.M.-C.); julucarli@gmail.com (J.L.C.-L.); 3Molecular Image Unit, Radiopharmacy, Fundación General de la Universidad de Málaga, 29010 Málaga, Spain; sfgarcia@fguma.es; 4Department of Medicine and Dermatology, University of Málaga, Instituto de Investigación Biomédica de Málaga (IBIMA), 29010 Málaga, Spain; psanchez@uma.es (P.S.-S.); valdivielso@uma.es (P.V.); 5Unit of Angiology and Vascular Surgery, Hospital Quirón, 29004 Málaga, Spain; arodriguezmorata@gmail.com; 6Unit of Ophtalmology, Hospital Universitario Virgen de la Victoria, 29010 Málaga, Spain; Imba5@hotmail.com; 7Department of Human Physiology, University of Málaga, and Instituto de Investigación Biomédica de Málaga (IBIMA), 29010 Málaga, Spain; igf@uma.es

**Keywords:** pseudoxanthoma elasticum, Phenodex, disease activity, 18F-NaF PET/CT

## Abstract

Pseudoxanthoma elasticum (PXE) is a genetic disease characterized by the calcification of elastin fibers. Our aim was to quantify vascular calcification in the arteries and the deposition of 18F-*sodium-fluoride* (18F-NaF) in the skin and vessel walls with positron emission tomography/computed tomography. This was an observational study including 18 patients with PXE. Vascular calcification was measured in Agatston units, and deposition in the skin and vessel walls was shown using target-to-background ratio (TBR). Severity of the disease was scored by Phenodex. We found higher vascular calcification in the popliteal, femoral, and aortic arch vessels compared to other vascular regions; however, the uptake of radiotracer was the highest in the aorta and femoral arteries. In the skin, the highest uptake was observed in the neck and the axillae. There was no significant association between 18F-NaF deposition in the arteries or skin and the global Phenodex score. In contrast, the Phenodex score was significantly associated in univariate analyses with the averaged vascular calcium score (*p* < 0.01). In the neck, patients with higher skin Phenodex scores exhibited higher radiotracer uptake. As a conclusion, because vascular calcification is physiological, our data suggested that the detection of cutaneous (neck) 18F-NaF deposits might serve to monitor the calcification process in the short-term for patients with PXE.

## 1. Introduction

Pseudoxanthoma elasticum (PXE) is a rare disease caused by mutations in the ABCC6 gene, which encodes multidrug resistance-associated protein 6 (MRP6). MRP6 is highly expressed in the liver and the kidney. Patients with PXE have less ATP available for conversion into inorganic pyrophosphate (PPi), a physiological inhibitor of calcification [1].

We [2] and others [3] have shown that serum PPi is reduced by a third in patients with PXE compared to controls. Consequently, PXE is characterized by progressive calcification of elastic fibers, which then undergo fragmentation and deformation. This condition leads to loss of function and damage to Bruch’s membrane in the retina, the skin, and vessel walls. The severity of PXE is evaluated with the Phenodex score [4]. This score rates the signs and symptoms in different systems (skin, eye, gastrointestinal system, vasculature, and the heart), and 12 is the highest score.

To date, no disease-modifying therapy is available that can slow or arrest PXE disease evolution. Furthermore, because PXE is a slowly progressing disease that takes many years to develop, it is difficult to assess the clinical benefit of any potential treatment. In one clinical trial, serial skin biopsies were used to quantify the calcium deposits [5].

Lack of PPi in PXE contributes to the formation of hydroxyapatite in tissues rich in elastic fibers (skin, eye, vessel wall) [6,7]. The radiotracer 18F-NaF binds to new deposits of hydroxyapatite [8]. This property provides an opportunity to identify and quantify dystrophic micro-calcifications. Moreover, when 18F-NaF was previously used with positron emission tomography/computed tomography (PET/CT) in subjects with PXE, it was shown to detect calcifications in vessel walls and the skin [9,10]; however, in a recent trial, femoral 18F-NaF uptake measurements did not show any change during a 12-month follow-up [11].

Thus, 18F-NaF is an imaging tool that allows for the assessment of active calcification processes in contrast to Computer Tomomgraphy (CT) that evidence calcification. Our aim was to quantify vascular calcification in the arteries and the deposition of 18F-NaF in the skin and vessel walls with positron emission tomography/computed tomography.

## 2. Experimental Section

Patients were recruited through an advertisement to the National Spanish Association for PXE from 1 December 2017 to 7 May 2018. Patients were eligible when they fulfilled the clinical and molecular criteria for the diagnosis of PXE (typical skin and eye symptoms plus two pathogenic mutations in the ABCC6 gene) [12]. Of the 25 patients selected, 18 provided signed informed consent. The Ethical Review Committee of Málaga approved the study on 15 November 2015. The study was adherent to the Declaration of Helsinki.

We recorded data on patient age, sex, anthropometrics, comorbidities, and therapies. Intermittent claudication was assessed with the Edinburg questionnaire [13]. Blood samples were drawn after an overnight fast. Hematological and biochemical parameters were measured using a Siemens Dimension Vista System Flex^®^ apparatus (HealtheCare GmbH, Erlangen, Germany). Before the scan, each patient was reviewed by an experienced ophthalmologist, dermatologist, and vascular specialist in the PXE field. The severity of the disease was determined with Phenodex score, which is the sum of organs affected by the disease (skin, eye, heart, and vascular), with 12 being the highest score possible [4]. For the purpose of this study, we performed extra-detailed analyses of the skin at the neck, axillae, elbow, and popliteal areas on both sides of the body, because although affectation is bilateral it is not symmetrical. These clinical analyses were evaluated on a scale of 0 to 3 points according to Phenodex score: 0 = no skin lesion; 1 = papules/bumps; 2 = plaques of coalesced papules; and 3 = lax and redundant skin ([4]

Fluorine-18 was produced with a GE PETtrace cyclotron (General Electric, Uppsala, Sweden). The 18F-NaF radiotracer was synthesized in a Tracerlab FXFN platform (GE Healthcare, Uppsala, Sweden). Activity was initially trapped in an anion-exchange cartridge (Light Accell QMA Accell, Waters Corporation, Miford, MA, USA) that had been conditioned with 10 mL of ethanol and 10 mL of water. The cartridge was then washed with 10 mL of sterile water, and the 18F-NaF activity was eluted from the QMA (Water Corporation, Milford, MA, USA) with 6 mL of sterile 0.9 % saline. Finally, the bulk solution of radiopharmaceutical was passed through a 0.22-µm filter (Millex GS, Merck Millipore, Burlington, MA, USA) into a sterile vial under aseptic conditions. All radiopharmaceutical quality control assays were performed as described previously [14].

Images were acquired with a PET/CT (General Electric Discovery ST, Chicago, Ill, USA) at the Molecular Unit of the Centro de Investigaciones Médico-Sanitarias (CIMES), Fundación General de la Universidad de Málaga, Universidad de Málaga. All participants received 3.7 MBq/kg of 18F-NaF (maximal dose 370 MBq). A whole-body PET scan was performed in all patients 90 min after the radiotracer injection, and the protocol did not change along the study period. The acquisition protocol included a low-dose CT study (autoMA, 140 kV, 3.75-mm slice thickness, pitch 1.5, 0.8 s/rot) for attenuation correction, anatomic location, and calcification evaluation, with the patient in supine position with the arms down, followed by a 3D-mode PET image with matrix size of 128×128 pixels and acquisition time of 3-4 min per bed position. Data were reconstructed with the iterative reconstruction algorithm (VUE Point HD, GE Healthcare, Uppsala Sweden) implemented in the scanner software. The scanner has European Association Nuclear Medicine Research Ltd. (EARL) accreditation for 18F-FDG PET/CT [15].

We measured 18F-NaF uptake and calcium deposits in vascular territories (left and right carotid arteries, ascending, arch, descending, and abdominal aortas, and right and left iliac, femoral, and popliteal arteries). The 18F-NaF uptake was also measured in skinfolds of the neck, axillae, elbow, groin, and popliteal areas on both sides. For each territory, we recorded the maximal and mean standard uptake value (SUV). The co-registered PET/CT images were taken as a basis on which circular region of interest (ROIs) were drawn that covered the wall and lumen of the measured arteries. ROIs of 1 cm in diameter were drawn on the higher caliber arteries, and their course was checked to locate the point with the highest maximum SUV (SUVmax) value. Vascular measurements were performed according to methods validated for 18F-FDG with the active (most diseased) segments approach, recording the maximum value (SUVmax) and mean value (SUVmean) from three consecutive segments including maximum uptake [16]. We adjusted the maximal and mean SUVs with the mean values of three measurements taken at the level of the right atrium [17], as described in the literature for 18F-NaF [18]. Values shown as final results are expressed as the target-to-background ratio (TBR, maximal and mean).

For skin measurements, volumes of interest (10-mm-diameter VOIs) were used in each region, recording SUVmax and SUVmean of isocontours with a threshold of 50% from the maximum value. The threshold was established by averaging the measurements of five patients in lumbar skin, avoiding subcutaneous fat inclusion. SUVs were adjusted with the mean values of the three measurements taken in the lumbar skin. Lumbar skin was chosen because is rarely affected in PXE [19]. Final results are expressed as the target-to-background ratio (TBR, maximal and mean). Vascular calcification was evaluated on a Carestream VUE Motion platform (CareStream Health, Inc, Onex, Toronto, Canada); data are expressed as the Agatston calcium score [20], volume, and mass.

Statistical analyses were performed with SPSS 25.0 (IBM SPPS Statistics, Chicago, IL, USA). Parameters are expressed as the number (%), mean ± SD, or median (interquartile range). For comparisons between groups, we used the χ2, Mann–Whitney, or Kruskal–Wallis tests. We considered a *p*-value < 0.05 significant. Lineal regression analyses were performed taking averaged vascular (Vasc) TBRmax, averaged Skin TBRmax, and averaged calcium score (dependent variables), as well as Phenodex score and age as covariates.

## 3. Results

### 3.1. General Data

This study included 18 patients with PXE, (13 women) aged 47 ± 13 years old. Of these, five patients were past smokers and two were current smokers; six had high-blood pressure and five had nephrolithiasis. None had diabetes. All patients with hypertension were under treatment with anti-hypertensive drugs; eight patients were under treatment with statins; and five were under treatment with antiplatelet agents. None of the patients were taking calcium analogues. Three patients had vascular disease; of these, one had stable angina pectoris, one had suffered an ischemic stroke, and one had stable angina pectoris with percutaneous revascularization. Five patients reported typical intermittent claudication. All general data are summarized by area in Appendix A, showing that the skin and eyes were most affected by the disease. PXE rarely affected the gastrointestinal (0 patients) and cardiac systems (2 patients).

### 3.2. Skin and Vascular Uptake of 18F-NaF

Table 1 shows the skin uptake of 18F-NaF, expressed as the TBRmax and TBRmean, for each skin area. The highest uptakes were recorded in the neck and the axillae; both uptakes were above the average of 10 sites. The average TBRmax and TBRmean values of the skin and vessels were not associated with the Phenodex score (range 2–9), based on either the Kruskal–Wallis or Spearman’s Rho test (*p* > 0.05).

In contrast, there was a weak association between the Phenodex score and the averaged calcium score (Kruskal–Wallis test, *p* = 0.088; Spearman’s Rho test 0.753, *p* < 0.01); however the association was lost when the analyses was adjusted by age. When patients were grouped according to the median Phenodex value (5.5), the averaged calcium score was significantly higher in those above the median than in those below the median (*p* < 0.01). However, the averaged vascular TBRmax and TBRmean only tended to be higher in patients above the median; and the averaged skin TBRs were not associated with the Phenodex scores (Table 2). In fact, using lineal regression analyses, Phenodex was not associated with averaged Vasc TBRmax, averaged Skin TBRmax, and averaged calcium score.

We evaluated the association between the averaged skin and vascular TBRs and the average calcium score according to the stages of severity (Phenodex levels) in the skin, eye, and vasculature. Again, we observed a non-significant trend, where the vascular TBRs tended to increase as the skin and the eye scores increased. However, this trend was not observed for the averaged skin TBRs. In contrast, for vascular disease, higher scores were associated with higher averaged calcium scores (*p* < 0.01). See detailed data in Appendix A No association was observed between the averaged TBRs and the presence/absence of nephrolithiasis and between the uptake in the skin and vessel wall (Pearson’s R – 0.322, *p* = 0.192)

When we examined specific skin sites, we observed that the axillae and the neck had the highest 18F-NaF uptake on both sides (see Appendix A) Moreover, for all sites except the axillae, on both sides we observed a trend, where the TBRmax and TBRmean values increased as the severity scores increased. Figure 1 and Figure 2 show 18F-NaF deposits at skin and vessel wall.

### 3.3. Vascular Calcium Score

Table 3 summarizes the deposition of calcium in the vascular territories explored. Remarkably, the most affected areas were the popliteal, femoral, and the aortic arch vessels, with calcium on the vessel wall in 12, 10, and 10 patients, respectively.

Table 4 shows the 18F-NaF uptake in vascular areas; the maximum and mean target-to-background ratios (TBRmax and TBRmean) were the highest in the aorta and femoral vessels.

When we analyzed the vascular TBRmax and TBRmean for each territory, according to the presence or absence of calcium, we observed that the highest uptake was recorded in the calcified areas; statistical significance was observed in the right iliac and both popliteal arteries. See Appendix A

## 4. Discussion

Calcifications of elastic fibers in Bruch’s membrane, arteries, and skin are hallmarks of PXE. Our study confirmed that 18F-NaF was deposited in calcified areas of the skin and vessel walls in PXE, but it was only weakly associated with disease severity.

Vascular calcification is not a rare condition; it is associated with arteriosclerosis, diabetes, and chronic kidney disease. In those conditions, it is age-dependent; and it affects most of the vasculature, extending to the coronary tree and to the thoracic and abdominal aorta. In contrast, in PXE, most calcification is located in the peripheral arteries of the lower extremities, and it is not related to the main vascular risk factors. In fact, in our series, the popliteal and femoral areas were the most frequently calcified areas; 12 out of 18 patients had calcification in this territory and these patients showed the highest calcium scores. Our results were consistent with those published recently, where a higher prevalence of vascular calcification was found in the femoro-popliteal arteries in subjects with PXE compared to controls, particularly in subjects younger than 55 years of age [21]. A combination of abnormal elastic fiber production by fibroblasts and a pro-calcifying environment [22] might be responsible for this high prevalence of peripheral vascular calcification in PXE [23]. Nonetheless, calcification can be found in up to 15% of subjects under 50 years of age without PXE, according to a recent pathological study [24].

As expected, both the skin and the vessel walls could take up 18F-NaF, and skin uptake was almost double the uptake in arteries. Concerning the skin sites, the maximal and mean TBR uptakes took place at the neck and the axillae, which is not surprising because the neck and axillae are the first places where skin anomalies are observed, in many cases during adolescence. At all sites except the axillae we observed a weak, non-significant, positive correlation between clinical severity and the TBRmax and TBRmean. At the axillae, we believe that the body position during the scan (arms-down) might have influenced the 18F-NaF measurements, which led to an overestimation of the measurements. Among the arteries, maximal 18F-NaF uptake was measured in the abdominal aorta and the femoral arteries. Interestingly, in arteries with calcification, 18F-NaF uptake tended to be higher than in those without calcification; this difference reached statistical significance in the popliteal arteries, which is highly characteristic of PXE.

Our study failed to show a significant association between the averaged TBRmax and TBRmean in either skin or arteries, and the clinical disease severity assessed with the Phenodex score, even when nephrolithiasis was added. In contrast, averaged vascular calcification was clearly associated with disease severity. This was not surprising, because the higher calcification the higher long-term disease.

As mentioned above, 18F-NaF is adsorbed to calcified deposits with high affinity and specificity [8]. In addition, it has been shown that the 18F-NaF uptake is strongly correlated with calcium scores [25]. These features suggested that 18F-NaF detection with PET/CT might potentially serve as a surrogate end-point in clinical trials for patients with PXE, avoiding repeated skin biopsies.

Our data suggested that the best site to monitor might be the neck, due to the following findings: (1) Neck skin took up more 18F-NaF than any vessel wall; (2) the uptake of 18F-NaF in the neck tended to increase as the disease severity increased (this trend probably did not reach statistical significance due to low statistical power); (3) vascular calcification is a very common process, and thus it is not specifically indicative of PXE; (4) 18F-NaF uptake occurs physiologically in the arteries but not in skin [26]; and (5) at one year, etidronate was shown to reduce ectopic arterial mineralization by 4% in patients with PXE, despite a 6% increase in the TBR at the femoral level, in both treated and control groups [11].

The main limitation of this study was the small number of patients included because it was a single center study. There was also a lack of a control group. Despite these limitations, our data confirmed the cutaneous deposition of 18F-NaF, especially at the neck level.

## 5. Conclusions

This study suggests that the quantification of cutaneous ^18^F-NaF deposits, especially at the neck level, might serve to monitor calcium deposition for patients with PXE in short-term studies.

## Figures and Tables

**Figure 1 jcm-09-01393-f001:**
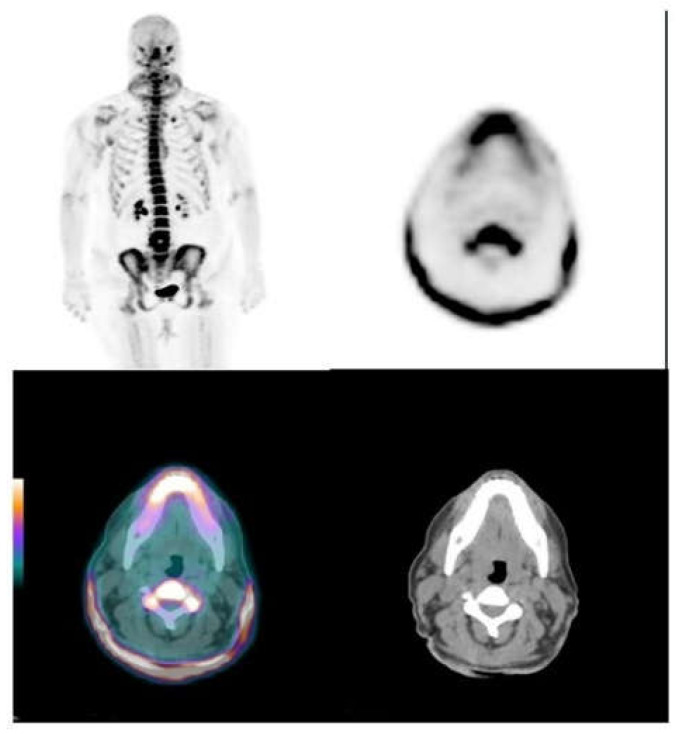
The ^18^F-NaF neck skin uptake. Maximal intensity projections (MIP) and axial images with intense neck skin uptake in a pseudoxanthoma elasticum (PXE) patient with Phenodex index of 6 (left panel).

**Figure 2 jcm-09-01393-f002:**
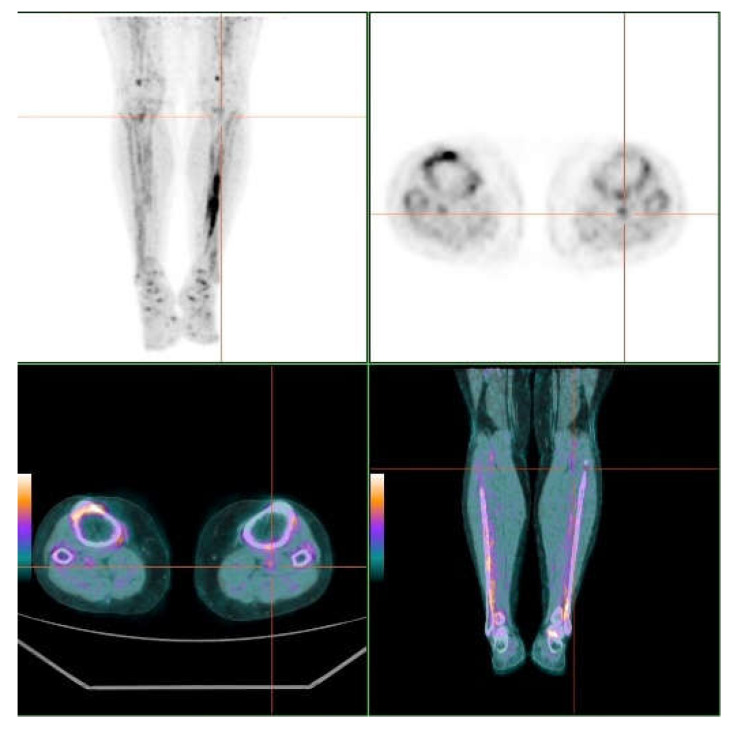
MIP and axial images at the groin level. Popliteal arterial uptake of ^18^F-NaF associated to calcification in the vessel wall. Uptake in the interosseus membrane of the left leg (right panel).

**Table 1 jcm-09-01393-t001:** Uptake of 18F-NaF by skin territories.

Territory	TBRmax	TBRmean
Lumbar skin SUV	0.54 (0.47–0.81)	0.39 (0.31–0.54)
Neck		
Right	5.91 (3.68–13.39)	4.14 (2.41–9.43)
Left	6.67 (3.32–13.4)	3.53 (2.20–8.96)
Axillae		
Right	5.89 (3.47–9.93)	3.67 (2.30–6.11)
Left	5.00 (3.13–9.97)	3.19 (2.00–3,35)
Elbow		
Right	2.32 (1.08–4.45)	1.72 (0.97–3.35)
Left	1.86 (0.99–5.35)	1.44 (0.70–3.67)
Groin		
Right	4.09 (1.42–6.14)	2.93 (1.15–4.04)
Left	2.08 (1.47–5.69)	1.54 (1.09–4.10)
Popliteal		
Right	3.24 (2.52–4.93)	2.16 (1.56–3.51)
Left	3.21 (2.19–4.89)	2.10 (1.56–3.51)
AVERAGE	4.22 (2.44–7.64)	2.79 (1.71–5.32)

Data shown as median (IQR). TBR: target-to-background ratio; SUV: standard uptake value.

**Table 2 jcm-09-01393-t002:** Averaged 18F-NaF TBRmax and TBRmean in vessels and skin according to the Phenodex median.

	PHENODEX		
	< p50	> p50	Sig
Av Vasc TBRmax	2.03 (1.80–275)	2.60 (2.24–2.84)	NS
Av Vasc TBRmean	1.73 (1.62–2.32)	2.16 (1.74–2.37)	NS
Av Skin TBRmax	4.38 (2.29–9.30)	4.07 (2.92-–6.38)	NS
Av Skin TBRmean	2.64 (1.52–6.23)	2.94 (2.05–4.63)	NS
Av calcium score	1.50 (0.25–41)	301 (159–541)	*p* < 0.05

Sig = p value. NS = non-significant. AV = averaged. Data shown as median (IQR).

**Table 3 jcm-09-01393-t003:** Deposition of calcium among arterial territories explored.

Territory	Number	Calcium Score	Calcium Volume	Calcium Mass
Carotid	7			
Right		0 (0–33,6)	0 (0–29,0)	0 (0–18,4)
Left		0 (0–18.4)	0 (0–20,7)	0 (0–3,4)
Ascending aorta	3	0 (0–0)	0 (0–0)	0 (0–0)
Aortic arch	10	9 (0–54)	24 (0–75)	2.9 (0–12)
Descending aorta	6	0 (0–168)	0 (0–186)	0 (0–33)
Abdominal aorta	7	38 (0–447)	36 (0–409)	7.8 (0–105)
Iliac	8			
Right		0 (0–517)	0 (0–436)	0 (0–107)
Left		0 (0–279)	0 (0–236)	0 (0–62)
Femoral	10			
Right		101 (0–684)	137 (0–680)	22 (0–143)
Left		82 (0–732)	70 (0–693)	14 (0–136)
Popliteal	12			
Right		20 (0–493)	43 (0.775)	6 (0–116)
Left		21 (0–489)	40 (0–745)	5 (0–112)

**Table 4 jcm-09-01393-t004:** TBRmax and TBRmean among arterial territories explored by the uptake of 18F-NaF PET/CT.

Territory	TBRmax	TBRmean
SUV/cava	0.88 (0.72–0.98)	0.67 (0.48–0.76)
Carotid		
Right	1,74 (1.43–2.19)	1,58 (1.27–1.97)
Left	1.83 (1.53–2.35)	1.64 (1.29-1.96)
Aorta		
Ascending	1.60 (1.41–1.89)	1.26 (1.15–1.50)
Arch	2.16 (1.72–2.34)	1.69 (1.40–1.87)
Descending	2.79 (2.19–3.22)	2.21 (1.78–2.85)
Abdominal	3.88 (3.36–4.64)	3.31 (2.79–4.04)
Iliac		
Right	2.11 (1.54–2.92)	1.72 (1.04–2.30)
Left	1.93 (1.40–2.40)	1.59 (1.20–2.01)
Femoral		
Right	2.83 (2.44–3.34)	2.23 (1.84–2.86)
Left	2.88 (2.26–3.70)	2.34 (1.88–2.66)
Popliteal		
Right	1.73 (1.15–2.20)	1.48 (0.98–1.74)
Left	2.00 (1.31–2.60)	1.52 (1.10–2.21)
AVERAGED	2.53 (2.00–2.78)	2.00 (1.64–2.32)

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
