# Peer review of "Skin and Arterial Wall Deposits of 18F-NaF and Severity of Disease in Patients with Pseudoxanthoma Elasticum"

_jcm, 2020, doi:10.3390/jcm9051393_

Round 1

Reviewer 1 Report

This is an observational study that aims to quantify the deposition of 18F-NaF in the skin and vessel walls of Pseudoxanthoma Elasticum (PXE) Patients using positron emission tomography/computed tomography. The study included 18 PXE patients, of which 3 patients had vascular disease including stable angina pectoris, ischemic stroke, and even underwent percutaneous revascularization.

Previous studies have already showed cutaneous deposition of 18F-NaF, especially at the neck and vessel wall. The current study does not add much to the literature, but rather confirms that 18F-NaF is deposited in calcified areas of the skin and vessel walls in PXE. They show that 18F-NaF deposition was not significantly association with disease severity, but rather the vascular calcium score was.

Therefore, the title of the manuscript is misleading, and the authors should change it to better reflect the results.

In summary, the small number of patients, the heterogeneity of the patients and the lack of control group damper the impact of the study.

The manuscript contains several typos and needs to be edited.

The quality of the figures is low.

Author Response

REVIEWER 1

This is an observational study that aims to quantify the deposition of 18F-NaF in the skin and vessel walls of Pseudoxanthoma Elasticum (PXE) Patients using positron emission tomography/computed tomography. The study included 18 PXE patients, of which 3 patients had vascular disease including stable angina pectoris, ischemic stroke, and even underwent percutaneous revascularization.

Previous studies have already showed cutaneous deposition of 18F-NaF, especially at the neck and vessel wall. We agree with the referee that the skin deposit of 18F-NaF has been described in the literature; only 4 patients in the paper of Mention et al (Mention et al., 2018) and 3 in the paper of Oudkerk et al (Oudkerk et al., 2016). The analyses were not so systematic and did not included all the areas which are clinically relevant for the disease.  The current study does not add much to the literature, but rather confirms that 18F-NaF is deposited in calcified areas of the skin and vessel walls in PXE. They show that 18F-NaF deposition was not significantly association with disease severity, but rather the vascular calcium score was. Our aim was to add new knowledge on the deposition of 18F-NaF in the vessel wall and in the skin and relate the findings with the severity of the disease. We agree with the reviewer that only vascular calcium score was associated with the severity; it make sense, because vascular calcification as well the severity of PXE measured by Phenodex is age dependent. In contrast, 18F-NaF deposition on the skin is an active process (the current formation of calcification) which is abnormally high in PXE.

Therefore, the title of the manuscript is misleading, and the authors should change it to better reflect the results. We agree with the reviewer that our title may induce to believe that there is an association of 18F-NaF skin deposit and severity. We have change to this: “Skin and Arterial Wall Deposits of 18F-NaF and Severity of the Disease in Patients with Pseudoxanthoma Elasticum”

In summary, the small number of patients, the heterogeneity of the patients and the lack of control group damper the impact of the study. The small number of participants is a rule in rare diseases research, especially if it has been done in a single center and is also limited by the budget of the grant.

The manuscript contains several typos and needs to be edited. The manuscript has been reviewed and edited.

The quality of the figures is low. We have improved the quality of the figures.

Mention, P.-J., F. Lacoeuille, G. Leftheriotis, L. Martin, and L. Omarjee, 2018, 18F-Flurodeoxyglucose and 18F-Sodium Fluoride Positron Emission Tomography/Computed Tomography Imaging of Arterial and Cutaneous Alterations in Pseudoxanthoma Elasticum: Circulation: Cardiovascular Imaging, v. 11, no. 1, doi:10.1161/CIRCIMAGING.117.007060.

Oudkerk, S. F., P. A. de Jong, B. A. Blomberg, A. M. Scholtens, W. P. T. M. Mali, and W. Spiering, 2016, Whole-Body Visualization of Ectopic Bone Formation of Arteries and Skin in Pseudoxanthoma Elasticum: JACC: Cardiovascular Imaging, v. 9, no. 6, p. 755–756, doi:10.1016/j.jcmg.2015.06.006.

Reviewer 2 Report

In this article, the authors investigated 18F Sodium Fluoride (18F NaF) uptake the arteries and the skin of patients with Pseudoxanthoma elasticum (PXE) and investigate the relation of 18F NaF PET/CT uptake and disease severity based on the Phenodex scale. PXE is a monogenetic disorder which results in extensive calcification of the arteries and the skin. 18F NaF PET/CT is a novel method to quantify (micro) calcifications and, as the authors argue, might be an intermediate endpoint for clinical trials into PXE. This is the first article to quantify 18F NaF PET uptake in both the skin and several arterial beds in PXE, which makes it an interesting addition to our knowledge on PXE. I do however have some major and minor issues that need to be addressed before it can be accepted for publication.

Issues

In general:

  1. The TBR skin measurements are not validated. Could the authors provide data on interrater agreement, reliability and repeatability?
  2. The authors perform a lot of statistical analyses on a small group of 18 PXE patients. Since the primary aim of this study is to describe the NaF PET profile in the skin and arteries of PXE patients, the authors should present the descriptive data on NaF PET uptake in the arteries and the skin and withhold from extensive statistical analyses. For example: In table 3, the authors test the association between TBR in the skin, arteries and arterial calcium score with phenodex score. In table 4 the authors test the associations between TBR in the skin, arteries and arterial calcium and the four separate parts of the phenodex score and in table 5 the authors test the association between TBR in the skin and clinical severity of the skin calcifications in four separate locations (neck, axilla, elbow and popliteal). in table 8 (page 9), the authors provide data on TBR in 12 different arteries and compare TBR between patients with and without calcification in the concerned arteries. It is not clear why the difference in arterial TBR between patients with and without calcification is compared, since this was not a stated research question. The large number of statistical analyses in such a small patient group increases the risk of false positive findings. I would suggest to limit the number of statistical analyses and just test the association between TBR/calcium score (as the outcome) and phenodex classification (as the determinant) using linear regression models (see below) and adjust for age, since PXE is a slowly progressive disease.
  1. Averaged TBR skin ~ phenodex score + age
  2. Averaged TBR vascular ~ phenodex score + age
  3. Averaged calcium score ~ phenodex score + age

Introduction:

  1. What do the authors mean with “ to date, no pathogenic therapy is available”? Maybe “no disease modifying therapy”?
  2. The referencing does not seem accurate. E.g.
    1. line 70-71 “ however … follow-up” should be reference 9 instead of 11.
    2. line 78-80 “ Patients … ABCC6 gene” should be reference 10 instead of 12.

Methods

  1. Page 4, line 54. Clinical analyses of the skin was scored on a 4 point scale (0-3). It is not clear what scale is used and how the points are defined? Please provide definitions for this scale.
  2. Page 5, line 38-42: “We measured 18F Naf PET tracer uptake and calcium deposits in vascular territories on both sides of the body (carotids, ascending, arch, descending and abdominal aorta, iliac, femoral, and popliteal).” Please specify how many NaF PET slices were analyzed per vessel. Were all slices analyzed, or was it just one ROI?

Results

  1. Table 1: The Legend of this table seems missing, please provide a legend. Please change : Skin folders” in “ skin folds”. In addition, it is not clear what the scores mentioned under “ skin folds” mean.
  2. In table 2 the authors provide data on the TBR in the skin. It is not clear why TBR in the neck is divided into left and right. Since the aim of this study is to describe the NaF PET profile in PXE, please provide descriptive data on TBR per vessel bed in the carotid arteries (left and right together), the thoracic (ascending, arch, descending) aorta, the abdominal aorta, the iliac arteries (left and right) the femoral arteries (left and right) and the popliteal arteries (left and right) without division in subgroups. This can be added to table 2 which describes the 18f-NaF PET uptake in the skin.
  3. Table 7: please change PET/TC into PET/CT

Discussion

Page 10 line 226: “as expected, both the skin and the vessel walls could take up 18F NaF, and skin uptake was almost double the uptake in arteries.” This cannot be stated since the TBR of the arteries is calculated differently from the TBR of the skin.  

The authors conclude that cutaneous skin deposition of 18F NaF PET might be a surrogate endpoint for short-term clinical trials for patients with PXE but this conclusion is not supported by the data. Longitudinal follow up (as the authors plan to do) should provide data on the progression of NaF PET uptake in the skin and progression of disease. Despite extensive statistical analyses on the association between NaF PET uptake and clinical severity of the disease no such associations were found. Since the only significant association was found between CT measured vascular calcification and phenodex classification, might this be a proper intermediate endpoint for clinical trials?

Author Response

REVIEWER 2

In this article, the authors investigated 18F Sodium Fluoride (18F NaF) uptake the arteries and the skin of patients with Pseudoxanthoma elasticum (PXE) and investigate the relation of 18F NaF PET/CT uptake and disease severity based on the Phenodex scale. PXE is a monogenetic disorder which results in extensive calcification of the arteries and the skin. 18F NaF PET/CT is a novel method to quantify (micro) calcifications and, as the authors argue, might be an intermediate endpoint for clinical trials into PXE. This is the first article to quantify 18F NaF PET uptake in both the skin and several arterial beds in PXE, which makes it an interesting addition to our knowledge on PXE. I do however have some major and minor issues that need to be addressed before it can be accepted for publication.

Issues

In general:

The TBR skin measurements are not validated. Could the authors provide data on interrater agreement, reliability and repeatability? Due to the geometrical characteristics of skin deposits is difficult to standardize regions of interest tracing in comparison with vascular uptake. Measurements have been done following regular standards used in nuclear medicine and particularly in PET-CT. Semiquantitative values and indexes respect reference regions. We agree that validation in a method only published in a pair of short communications is of great interest but due to the insuficient number of patients scanned we have planned a larger research when scans from three year after initiation of treatment would be available.

  1. The authors perform a lot of statistical analyses on a small group of 18 PXE patients. Since the primary aim of this study is to describe the NaF PET profile in the skin and arteries of PXE patients, the authors should present the descriptive data on NaF PET uptake in the arteries and the skin and withhold from extensive statistical analyses. For example: In table 3, the authors test the association between TBR in the skin, arteries and arterial calcium score with phenodex score. In table 4 the authors test the associations between TBR in the skin, arteries and arterial calcium and the four separate parts of the phenodex score and in table 5 the authors test the association between TBR in the skin and clinical severity of the skin calcifications in four separate locations (neck, axilla, elbow and popliteal). in table 8 (page 9), the authors provide data on TBR in 12 different arteries and compare TBR between patients with and without calcification in the concerned arteries. It is not clear why the difference in arterial TBR between patients with and without calcification is compared, since this was not a stated research question. The large number of statistical analyses in such a small patient group increases the risk of false positive findings. I would suggest to limit the number of statistical analyses and just test the association between TBR/calcium score (as the outcome) and phenodex classification (as the determinant) using linear regression models (see below) and adjust for age, since PXE is a slowly progressive disease. We agree that our results are very profuse, and they can be shortened. Some of the Tables have been added to a supplementary file.
  1. Averaged TBR skin ~ phenodex score + age
  2. Averaged TBR vascular ~ phenodex score + age
  3. Averaged calcium score ~ phenodex score + age

We re-analyzed data according to the proposal. The results are in the Table

AveragedTBR vascular

Non adjusted

Adjusted by age

Standardized β coefficient

p value

Standardized β coefficient

p value

Phenodex

0.47 (0.01-0.233)

0.048

0.326 (-0.06-0.221)

NS

Averaged TBR skin

Phenodex

-0.011 (-0.964 – 0.928)

NS

0.326 (0.447-1.690)

NS

Averaged Calcium Score

Phenodex

0.427 (-16.09-166.4)

0.09

0.233 (-64.4 – 146.6)

As it is seen, Phenodex was only associated with averaged TBR vascular but the association was lost after adjustments was done for age. We believe this data do not change the result or the conclusion of our paper. We have included a short sentence in results (see paragraph from 160 to 169 line in revised manuscript).

Introduction: 

  1. What do the authors mean with “ to date, no pathogenic therapy is available”? Maybe “no disease modifying therapy”? Absolutely, the sentence suggested by the reviewer is more accurate. We have changed in the text.
  2. The referencing does not seem accurate. E.g.
    1. line 70-71 “ however … follow-up” should be reference 9 instead of 11.
    2. line 78-80 “ Patients … ABCC6 gene” should be reference 10 instead of 12.

We really appreciate the comments, because some of the references were mistakenly cited. We have reviewed all of them.

Methods 

  1. Page 4, line 54. Clinical analyses of the skin was scored on a 4 point scale (0-3). It is not clear what scale is used and how the points are defined? Please provide definitions for this scale. This sentence has been added to the text: “, according to Phenodex Score: 0 = no skin lesion; 1 = papules/bumps; 2 = Plaques of coalesced papules and 3 =  lax and redundant skin (Pfendner et al., 2007)

  1. Page 5, line 38-42: “We measured 18F Naf PET tracer uptake and calcium deposits in vascular territories on both sides of the body (carotids, ascending, arch, descending and abdominal aorta, iliac, femoral, and popliteal).” Please specify how many NaF PET slices were analyzed per vessel. Were all slices analyzed, or was it just one ROI?. This sentence from the same paragraph has been modified: Vascular measurements were performed according to methods validated for 18F-FDG with active (most diseased) segments approach recording maximum value (SUVmax) and mean value (SUVmean) from three consecutive segments including maximum uptake”.

Results

  1. Table 1: The Legend of this table seems missing, please provide a legend. Please change: Skin folders” in “skin folds”. In addition, it is not clear what the scores mentioned under “skin folds” mean. Legend of the Table has been added and the scores have been clarified.
  2. In table 2 the authors provide data on the TBR in the skin. It is not clear why TBR in the neck is divided into left and right. Since the aim of this study is to describe the NaF PET profile in PXE, please provide descriptive data on TBR per vessel bed in the carotid arteries (left and right together), the thoracic (ascending, arch, descending) aorta, the abdominal aorta, the iliac arteries (left and right) the femoral arteries (left and right) and the popliteal arteries (left and right) without division in subgroups.  This can be added to table 2 which describes the 18f-NaF PET uptake in the skin. The reason for the independent measures on the left and right side for arteries (and by extension to the skin folds) is that is the usual way in clinical practice. Even arteriosclerosis or vasculitis are systemics, it is very common to have a more severe lesion in one vessel or area. We believe that to sum up both side does not add extra value to our results.
  3. Table 7: please change PET/TC into PET/CT. Done.

Discussion

Page 10 line 226: “as expected, both the skin and the vessel walls could take up 18F NaF, and skin uptake was almost double the uptake in arteries.” This cannot be stated since the TBR of the arteries is calculated differently from the TBR of the skin.  We believe that our statement is correct. It is true that TBR is calculated differently in vessel wall and the skin; the background for the vessel was measurement taken at the right atrium (0.87 ± 0.18); for the skin, the measurement was in the lumbar skin (0.61 ± 0.19). But if we only take into account the SUVmax, the value recorded in the skin (neck and axillae) is almost the double also than SUVmax at the carotid and the thoracic aorta. Since both type of measurements are expected to reveal the relationship between affected tissue and a reference region devoid of calcification we think that TBRs may be comparable.

The authors conclude that cutaneous skin deposition of 18F NaF PET might be a surrogate endpoint for short-term clinical trials for patients with PXE but this conclusion is not supported by the data. Longitudinal follow up (as the authors plan to do) should provide data on the progression of NaF PET uptake in the skin and progression of disease. Despite extensive statistical analyses on the association between NaF PET uptake and clinical severity of the disease no such associations were found. Since the only significant association was found between CT measured vascular calcification and phenodex classification, might this be a proper intermediate endpoint for clinical trials? The reviewer raises a very important issue. For sure, vascular calcification is a proper intermediate endpoint for a clinical trial, but for medium or long-term trials because vascular calcification takes long time to develop. In contrast, 18F-NaF in the skin (or elsewhere) shows active deposition of hydroxyapatite crystals. Taking into account the pathogenesis of the disease (low serum PPi levels), drugs able to increase PPi (intake of soluble PPi, inhibitors of TNAP activity, and so on…) may reduce the uptake of 18F-NaF in the skin in the short term (months), indicating the potential benefit in the long term. By contrast, any drug for PXE not able to show a reduction in deposits of 18F-NaF probably will not have a beneficial impact in the disease. Furthermore, as mentioned in the manuscript, vascular calcification is a physiological process which is affected not only by age but also accelerated by common vascular risk factors; this is not the case for skin deposits of 18F-NaF.

Round 2

Reviewer 2 Report

I have no additional comments.